# Demonstration project of oral TDF-containing PrEP, administered, once-daily orally to men having sex with men (MSM) and transgender women (TGW) in India: Study protocol

**Seema Sahay**[1]*, **Sampada Bangar**[2], **Nomita Chandhiok**[3]

**1** ICMR-National AIDS Research Institute, Division of Social and Behavioral Sciences, Pune, India, **2** ICMR-National AIDS Research Institute, Division of Epidemiology and Biostatistics, Pune, India, **3** Indian Council of Medical Research, New Delhi, India

* ssahay@nariindia.org

## Abstract

### Background

India has successfully reduced Human Immuno-deficiency Virus (HIV) incidence, with a 66% decline since the year 2000 has been seen; however, control among Men having sex with Men (MSM) and transgender women (TGW) remains a critical challenge. Oral Pre-Exposure Prophylaxis (PrEP) may help close a critical HIV prevention gap for MSM and TGW in India; however, no studies to date have evaluated the feasibility of oral PrEP among MSM and TGW in India.

### Methods

The proposed study aims at understanding the implementation of the provision of daily oral-Tenofovir (TDF) containing PrEP among MSM and TGW through the clinic and community-based delivery models in Pune, Maharashtra, and Jalandhar, Punjab respectively in India. The study aims at estimating PrEP adherence, facilitators, and barriers to PrEP use, retention, acceptability, and willingness to pay for PrEP. After the screening, eligible participants (n = 600) will receive PrEP medicines and will be monitored quarterly for HIV, STIs, and renal and liver toxicity for 12 months as per the schedule of events. The primary outcomes of interest are PrEP acceptability, PrEP adherence, retention rate, adverse medical events, and sexual behavioural changes with PrEP use and breakthrough infections while on PrEP. The study will assess the feasibility of two service delivery models; however, the data from the two service delivery models will be analyzed independently and will not be compared for feasibility and other outcome indicators. The study has been initiated after obtaining appropriate regulatory approvals.

### Discussion

PrEP is efficacious in preventing HIV among high-risk population however there are scarce data on providing PrEP to MSM and TGW. The study will provide critical evidence to

**Data Availability Statement:** No datasets were generated or analysed during the current study.

The data will be available at director@nariindia.org after the study completion.

**Funding:** This study is funded by Indian Council of Medical Research, Government of India (Grant Number: 5/7/2/2016-RBMH).

**Competing interests:** The authors have declared that no competing interests exist.

**Abbreviations:** AE, Adverse Events; AEMC, Adverse Event Monitoring Committee; ART, Anti-Retro Viral Therapy; CBO, Community-Based Organization; CDSCO, Central Drugs Standard Control Organization; CT, Chlamydia Trachomatis; HBV, Hepatitis B Virus; HCV, Hepatitis C Virus; HIV, Human Immunodeficiency Virus; HSS, HIV Sentinel Surveillance; HSV-2, Herpes Simplex Virus-2; ICMR, Indian Council of Medical Research; MSM, Men Having Sex with Men; NARI, National AIDS Research Institute; NG, Neisseria Gonorrhea; NGO, Non-government Organization; NRA, National Regulatory Authority; PEP, Post-Exposure Prophylaxis; PrEP, Pre-Exposure Prophylaxis; PSC, Project Steering committee; PWID, People Who Inject Drugs; SAE, Serious Adverse Event; STI, Sexually Transmitted Infections; TDF/ FTC, Tenofovir/ Emtricitabine; TGW, Transgender Women.

programs and policymakers on the implementation of PrEP in a "real world" setting, among MSM and TGW in India including identifying populations that can benefit most from this additional HIV prevention intervention along with acceptable delivery strategies and means of support for adherence.

## Trial registration

Not applicable being a demonstration project. Efficacy is already proven.

## Background

In India, the Human Immunodeficiency Virus (HIV) epidemic is concentrated among the high-risk population including Men Having Sex with Men (MSM) and transgender women (TGW) population. The estimated National adult (15–49 years) HIV prevalence in India in 2019 was 0.22% (0.17%– 0.29%) [1]. India has demonstrated considerable success with a 37% decline in new HIV infections since the year 2010, yet around 69,000 new infections occurred in 2019 [1]. HIV prevalence among MSM and TGW recorded at the national level was 2.69% and 3.14% respectively [2]. Solomon et al in 2015 reported an estimated annual HIV incidence among MSM of 2.2% [3] and the estimated HIV incidence among MSM in Punjab is reportedly 0.14% [1]. Several pockets in the country have shown a higher HIV prevalence with mixed trends [4]. To monitor the HIV trend in diverse populations across the country, HIV Sentinel Surveillance (HSS) is being conducted every year in India [2]. In MSM, a decline in the HIV prevalence has been achieved from 7.41% in 2007 to 4.43% in 2011 and further to 4.3% in 2015 indicating plateauing of HIV prevalence in this population. The historical data on prevalence and trends among TGW is limited [2, 5]. In 2019, there were approximately 58.96 AIDS-related deaths that contributed to the declining HIV prevalence. However, to achieve End AIDS Epidemic goals, it is important to curtail the new infections in the high-risk population. Control of HIV among MSM and TGW remains a serious challenge due to the hidden nature and stigma associated with it. In the past few years, studies have demonstrated the efficacy of more potent combination ART-based regimes to prevent perinatal transmission, and ART initiation immediately after HIV diagnosis irrespective of CD4 levels for people living with HIV to prevent HIV transmission to serodiscordant partners [6]. The most recent addition to this list of proven prevention strategies offers HIV sexual transmission in high-risk populations through anal and/or vaginal intercourse regardless of partners' gender is the oral pre-exposure prophylaxis (PrEP) [7–9]. The U.S. Centers for Disease Control has recommended the targeted use of PrEP for MSM, heterosexual men and women, and people who inject drugs (PWIDs), who are at substantial risk of HIV [10]. Oral PrEP containing Tenofovir disoproxil fumarate (TDF) should be offered as an additional prevention choice for people at substantial risk of HIV infection as part of combination HIV prevention approaches [11]. In 2015, given the additional evidence of oral PrEP's effectiveness, the World Health Organzation (WHO) released a strong recommendation, based on high-quality evidence, and specifically recommended that oral TDF-containing PrEP should be offered to any person at substantial risk for HIV [12].

Oral PrEP may help in closing a critical HIV prevention gap for MSM and TGW in India. Currently, PrEP is not included in the National AIDS Control Program. However, for willing and affording MSM and TGW, private practitioners are prescribing PrEP medicines. Recently, National PrEP Technical guidelines have been released by the Government of India [13] The

goal of these guidelines is to provide correct information on PrEP and its uses for both providers and users. The usage of PrEP may also have additional benefits such as the uptake of opportunity for STI testing and therefore a potential for further STI prevention and control is visualized. In a willingness to use the PrEP study, it was shown that offering PrEP to men who test infrequently may serve to engage them more in routine HIV/STI testing and create a continued dialogue around sexual health between patient and provider to prevent HIV infection [14]. The effectiveness of uptake of PrEP use is dependent on the willingness of the community to use it which is reported to be high in various studies conducted among MSM if individual, social, and structural barriers are addressed [15–18]. According to a study on the acceptability of oral PrEP among Indian MSM, despite prior low awareness of PrEP, after gaining an understanding of the intervention, most MSM reported willingness to use PrEP [19]. Despite the recommendations of the World Health Organization for PrEP, and reported willingness, a gap between efficacy and effectiveness is predicted as many clinical trial researchers cited adherence as a major challenge for efficacy [11]. Therefore, the reported acceptability might not be a sufficient condition for the effectiveness of this intervention. No study to date has evaluated the feasibility of the use of oral PrEP by MSM and TGW in India. Many questions remain unanswered on how to implement PrEP in a "real world" setting in India, including identifying acceptable delivery strategies and means of support for adherence.

Since the identified population for oral PrEP are the key populations who might have a different worldview and may need different delivery models, we are conducting a demonstration project for oral PrEP among the key population of MSM and TGW persons in India. The study aims to assess whether oral PrEP can be added to a package of HIV prevention interventions in community settings and through facility-based delivery. The community-based approach for PrEP delivery could increase and sustain access to oral PrEP whereas it is envisaged that a facility-based delivery model could help in better monitoring for MSM and TGW persons in India. The purpose is to study the operational challenges in these two types of PrEP delivery settings. We specifically aim to study the acceptability, adherence, and safety of PrEP, and assess potential changes in sexual behaviour, and quality of life as a consequence of its use. The sustainability of PrEP also depends on its cost to the user. Sexual and reproductive health care in India is as low as Rs 11.43 per capita annually for the population in the reproductive age group including the population at risk of HIV which includes the direct cost of care and indirect cost of programs and systems [20]. Hence the current study is trying to explore the willingness of study participants to use PrEP if available at a subsidized cost under the program.

## Objectives

The study was designed following the WHO's PrEP demonstration projects: A framework for country-level protocol development [21]. The primary goal of the study is to evaluate the feasibility and potential impact of implementing HIV PrEP as part of combination prevention for HIV among MSM and TGW persons in India. The primary objectives are 1) Retention for study follow-up and its correlates among MSM and TGW persons receiving oral TDF containing PrEP, 2) Acceptability of provision of daily oral TDF containing PrEP among the study population through both community and clinic-based delivery models, 3) Estimation of adherence to Oral TDF-containing PrEP along with existing HIV prevention options in the country, 4) Usage of other HIV prevention options and change in risk perceptions and behaviour during or the absence of oral TDF-containing PrEP, 5) Evaluating change in sexual behaviours in response to PrEP among the study population, 6) Willingness to pay for PrEP at subsidized cost among study population and, 7) Understanding facilitators and barriers of PrEP use among the study population.

## Methods/design

This is a demonstration project to study the acceptability and PrEP administration in a real-world setting and not a clinical trial. Hence trial registration has not been done.

### Study setting

The HIV prevalence at MSM sites from the 15th round of HIV Sentinel Surveillance, 2017 was found to be 4.67 and 4.69 in the Indian states of Punjab and Maharashtra respectively which was higher than the National prevalence among MSM [2.69% (95% CI: 2.47–2.91)] [2] Hence these two states have been selected. The community-based setting will be established in Jalandhar, Punjab whereas, the facility-based setting will be established in Pune, Maharashtra. The facility-based setting is a well-equipped outpatient clinic existing on the campus of one of the government research organizations which was already catering to at-risk HIV-uninfected as well as HIV-infected populations. The clinic is providing HIV testing, counselling and risk reduction counselling to the attendees. This facility is selected because the key population is familiar with this clinic and the site of the clinic is well accessible from the railway station, bus stand and marketplaces. Community-based setting is established in an existing community-based organization which is also providing HIV-related services to at risk populations and it is run by the community. Both settings are manned by a team of clinicians, counsellors, field workers, nurse/lab technicians, and a pharmacist. We expect differences both in stigma and approach in the two regions not only because they are two geographic regions but also because one site is the facility and the other site is community-based. In the community-based setting the target population is visible and has the support of the community however in facility-based setting hidden MSM are expected. Hence there might be a difference in stigma in the two settings and the recruitment and implementation strategies might differ.

### Sample size

The sample size of 800 participants across two settings was finalized initially based on the WHO guidelines for conducting demonstration projects [21] wherein the roughly estimated sample size range is 600 to 800 for demonstration projects. While planning the study, we had chosen the upper limit of the range i.e., 800 sample size for both sites. However, the COVID pandemic posed challenges in the recruitment and retention of participants due to lockdowns, restricted travel, and adherence. To overcome these challenges and get enough power for outcomes, the sample size has been reduced to a lower range i.e., 600 (300 at each site) instead of 800. The final sample size is 600. Of these, 20% study population will be TGW population across both sites and will be analyzed separately for better understanding.

### Eligibility

The inclusion criteria include self-identified TGW person or MSM, aged 18 years, who are willing and able to provide written informed consent to participate. Participants are required to have a negative Hepatitis B Virus (HBV) and HIV test respectively within 7 working days before initiation of PrEP. Participants having a substantial HIV risk based on the last 6 months as per WHO's implementation tool for pre-exposure prophylaxis (PrEP) of HIV infection [22] for conducting demonstration projects will be considered eligible. The substantial risk is defined as an MSM having anal sexual intercourse or TGW person having vaginal or anal sexual intercourse without a condom with more than one partner, OR a sexual partner with one or more HIV risk factors, OR a history of a sexually transmitted infection (STI) by lab testing or self-report or syndromic STI treatment, OR use of post-exposure prophylaxis (PEP), OR

requesting PrEP, OR engaged in commercial sex work (goods/money/kind). The participant will be screened for HIV, HBV, Hepatitis C virus (HCV), no contraindication to oral TDF containing PrEP medications (TDF/FTC) [*HIV positive status*, *Estimated creatinine clearance less than 60mL/min and or weight less than 77 pounds for F/TDF*, *Estimated creatinine clearance less than 30 mL/min and or weight less than 77 pounds or having vaginal sex for F/TAF*, *unwillingness or inability to take pill every day or follow up every 3 months for lab work for F/TDF and F/TAF*, *sign and symptoms of acute HIV infection*, *allergy or contraindications to any medicine in the prep regimen*, *history of recent high-risk exposure*], and adequate renal function. The adequate renal function is ensured by the calculation of creatinine clearance $\geq$ 60 ml/min as estimated by the Cockcroft-Gault equation at screening.

The exclusion criteria include a participant with confirmed HIV, Hepatitis B Virus, and Hepatitis C virus-positive report. Other exclusion criteria included known or reported hypersensitivity to any medicine in the PrEP regimen (example): Tenofovir, and FTC or 3TC, currently taking PrEP or enrolled in any ART intervention project and at screening, has any other condition that, based on the opinion of the investigator or designee, would preclude the provision of informed consent; make participation in the project unsafe; complicate interpretation of outcome data, or otherwise interfere with achieving the project objectives.

## Enrollment /recruitment

Community preparedness and sensitization will be initiated in the study preparedness phase, focusing on identification and linkages. Group meetings with potential participants will be undertaken with the help of Community-Based Organizations (CBOs)/Non-Governmental Organizations (NGOs). Recruitment strategies also include posting advertisements at public places like bus stops, public toilets, traffic signals, community hotspots, and social media sites specific to the population of interest, conducting community meetings, approaching through social media, and using the snowball approach. Potential study participants who get identified in the community will be provided information about PrEP and referred to the local participating study setting for screening. Considering the sensitivity of the study population, the Pune site which is a facility-based site, will actively network with community-based organizations for community literacy, and support in recruitment and retention. Retention at both sites will be done as per the retention SOP.

## Study visits

Fig 1 describes the study implementation plan in brief.

At screening, the potential participant will be screened for HIV, HBV, HCV, liver and renal functioning, and confirm eligibility for enrollment. The samples received will be tested for HIV diagnosis following HIV Testing Strategy III defined by the National AIDS Control Organization (NACO) [23]. Three different kits with different principles of testing are used in this strategy. The samples are initially tested by the first HIV rapid kit. If the result is nonreactive, the sample is reported as HIV-negative. If the result of the sample is reactive it is subjected to two more HIV rapid tests. When the sample is HIV reactive by all three tests, it is reported as HIV positive. If there is a discrepancy in the results among three rapid tests, the status of the sample is confirmed by Western Blot. Participants will be assessed for acute HIV infection by a trained counsellor based on the history of unprotected sex in the past 4–6 weeks and for signs and symptoms of acute viral syndrome by the clinician at screening. If found in a window period based on history then he will be deferred enrollment for 3 months. Participants will be called for repeat HIV testing and enrolled based on repeat screening. HIV status will be

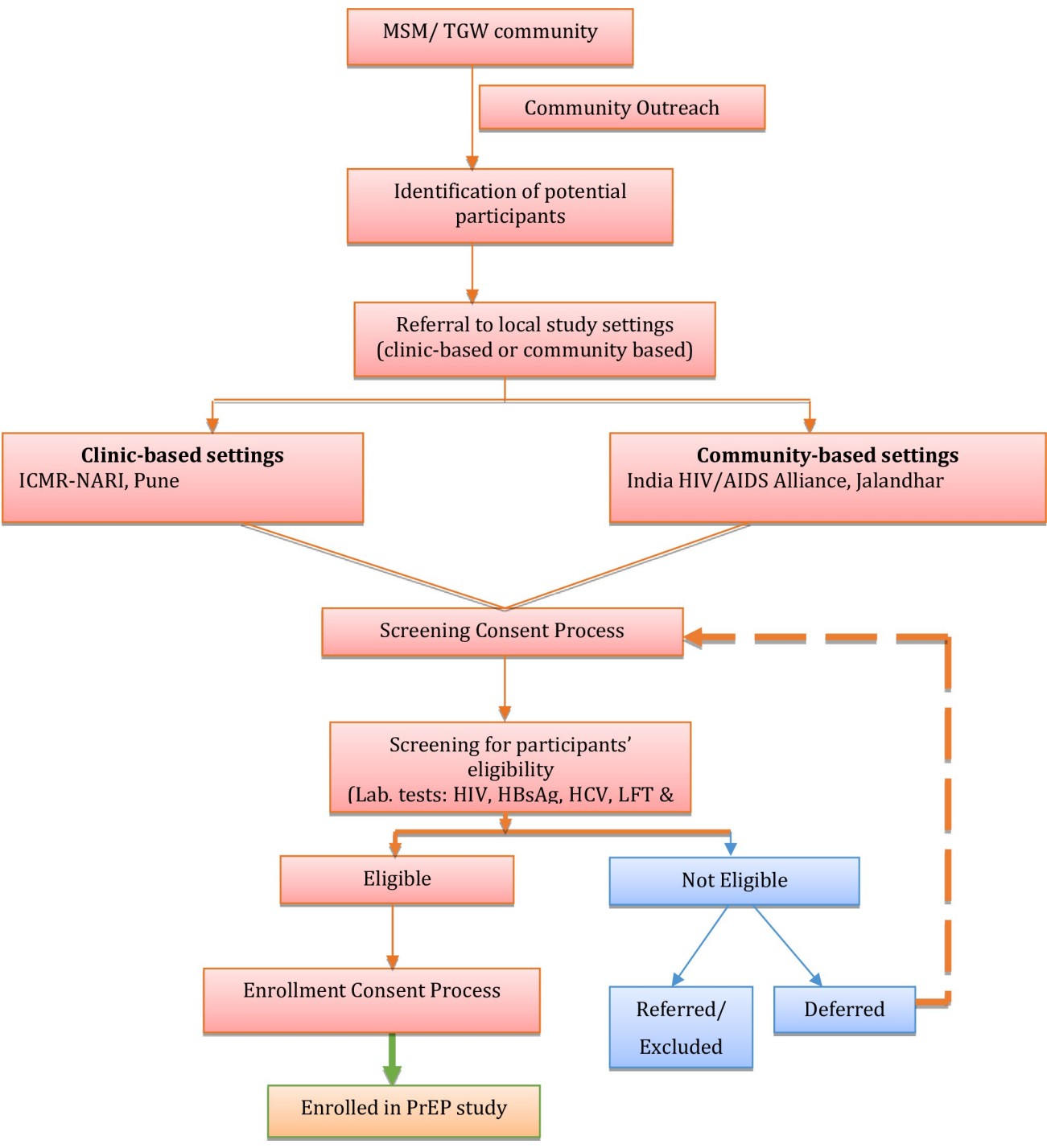

**Fig 1. Study implementation flow chart.**

confirmed using the *ARKRAY Healthcare Pvt Ltd. Comb Aids-RS HIV 1+2 immunodot test kit*, a 3rd generation HIV antibody test.

After the eligibility is confirmed, the participant will be called for enrollment within 7 days. At the enrollment visit (Visit 1), the trained counsellor will obtain consent for enrollment after explaining the purpose of the study, objectives, and the inclusion/ exclusion criteria to eligible

participants. The willing and consenting participants will be enrolled in the study. Baseline data will be collected, through a semi-structured questionnaire about their socio-demographic characteristics, sexual behaviour, substance use, and violence by a trained counsellor. The data on sexual behaviour will include the type and number of sexual acts and new sex partners, the use of a condom at each sexual act, any incidence of breakage or slippage of the condom if used in the last three months and symptoms of acute HIV infection. This data will be collected at baseline, 3, 6, 9, and 12 months and will help in assessing change in sexual behaviour due to PrEP use. We are collecting information on emotional distress, and abusive relations that could possibly lead to depression and suicidal ideations among participants. The exit interview would explore their experiences and emotions pre and post-use of PrEP.

PrEP medicines will be prescribed by the clinician and dispensed by a trained pharmacist along with adherence counselling. All enrolled participants will be given 3 doses of HBV vaccination (0, 1, and 6 months) starting preferably at the enrollment visit. Risk reduction counselling and information, and counselling on strategies to reduce HIV risk will be provided to the participants at each visit [24]. Follow-up study visits for creatinine monitoring and data collection for participants will be scheduled as per the schedule of events (Fig 2).

Participants will be followed up twice a month for the first three months and then every three months until 12 months. During the first six visits (fortnightly visits in the first three months), side effects will be assessed and adherence counselling will be done to meet the goal of oral PrEP becoming a routine practice for PrEP users. This will be continued at each visit to ensure optimal adherence. The blood sample will be collected at each visit for future Tenofovir level monitoring. TDF levels will be measured in all cases of seroconversion during the study and in 20% of randomly selected participants for adherence assessment. The blood sample will

**13. Appendices**
**Appendix A: Schedule of events**

| Study Visit | Informed Consent | Ongoing informed consent review | Data Collection | Blood collection | Genital/anal swab collection / Urine for NG CT PCR | HIV Testing | HBsAg testing | HSV | HCV | LFT | Sr creatine | STI screening (NG CT) | RPR for syphillis | Primary care STI Treatment | HBV Vaccination | Pills dispensing | Adherence counselling | Pill count | Storage of 2ml plasma sample |
|---|---|---|---|---|---|---|---|---|---|---|---|---|---|---|---|---|---|---|---|
| Screening | x | | x | x | | x | x | x | x | x | x | x* | | x | | | | | |
| Enrollment | x | x | x | x | x | | | x | | | | x | x | x | x | x | x | | x |
| 15 days | | | | | | | | | | | | | | | | | x | x | |
| 1 month | | x | x | | | | | | | | | | | x | x | x | x | x | |
| 1.5 month | | | | | | | | | | | | | | | | | x | x | |
| 2 month | | x | x | | | | | | | | | | | x | | x | x | x | |
| 2.5 month | | | | | | | | | | | | | | | | | x | x | |
| 3 month | | x | x | x | | x | | | | | x | x* | | x | | x | x | x | x |
| 6 month | | x | x | x | x | x | x | | x | x | x | x | x | x | x | x | x | x | x |
| 9 month | | x | x | x | | x | | | | | x | x* | | x | | x | x | x | x |
| 12 month | | x | x | x | x | x | x | x | x | x | x | X | x | x | | x | x | x | x |

12 months follow up for all study participants, termination visit at 12 month after the enrollment

 1 Data Collection will be done at all visits on ipads
 2 Blood will be collected for HIV testing screening 3,6, 9 and 12 month visit and in addition whenever any symptoms of acute HIV infection are observed or reported by the participants
 3 Participant will be screened for Syphillis, Neisseria Gonorrhea, Chalmydia Trachomatis at enrollment, 6 and 12th month visit and anal/genital swabs or urine samples will be collected.
 At screening, 3rd and 9th month visit there will be only clinical screening (*) which means clinician will assess the participants for symptoms of STIs and collect samples if needed only.
 HSV testing will be done at baseline and subsequent symptom driven testing
 4 Blood will be tested for TDF levels in case of suspected breakthrough HIV infection and for 20% of the samples.

**Fig 2. Schedule of events for the PrEP demonstration project.**

be collected at enrollment and quarterly thereafter. Plasma will be separated from the blood sample and will be stored in -20˚C freezer until they are processed. They will be processed using Liquid Chromatography Mass Spectrometry (LC-MS). Participants will be counselled about acute infection, condom use, and care for the regular partner, and linkage to care is being ensured. Rapid plasma reagin (RPR) on blood samples will be done to confirm the presence of antibodies to syphilis, Polymerase Chain (PCR) test will be done on urine samples for the direct, qualitative detection of the plasmid DNA for C. *trachomatis* and the genomic DNA of N. *gonorrhoea* at enrollment, 6 monthly and at termination study visit and also symptom-driven testing on the presence of ulcer will be done. HSV antibody test will be done for HSV-2 diagnosis at baseline (enrollment)

PrEP will be discontinued permanently

a. On completion of 12 months of follow-up on Oral TDF-containing PrEP

b. Acquisition of HIV infection

c. The participant expresses intent to leave the region or will be otherwise unavailable for follow-up

d. The participant voluntarily chooses to discontinue PrEP for disclosed or undisclosed personal reasons

e. Dose-limiting toxicity is found, requiring the discontinuation of PrEP, on request of the primary care provider if they feel that PrEP is no longer in the best interest of the participant

f. If the participant is judged by the investigator to be at significant risk of failing to comply with the provisions of the protocol so as to cause harm to self or seriously interfere with the validity of the results. Medication will be temporarily discontinued for any reactive HIV test, and permanently discontinued if HIV infection is confirmed.

## Study medication

All enrolled study participants will receive oral Tenofovir-containing PrEP tablets. Each tablet contains 300 mg of Tenofovir Disoproxil Fumarate (equivalent to 245 mg of Tenofovir Disoproxil or 136 mg of Tenofovir) and Emtricitabine (FTC) 300 mg. The tablets will be stored at 25˚C (77˚F), with excursions permitted to 15˚ to 30˚C (59˚ to 86˚F). Participants will be directed to take one tablet of TDF-containing PrEP orally daily with or without food. A 30 days supply of PrEP will be dispensed to the participants every month for the first three months. Subsequently, the dispensing will be done quarterly. Three such quarterly dispensing will be done.

## Stakeholders' engagement

The success of the study lies in the involvement of relevant stakeholders. During protocol development, the protocol was shared with experts from World Health Organization and others. The comments/suggestions given by experts, members of the Scientific Advisory Committee, Ethics Committee, and ICMR expert committee were incorporated into the protocol. The final protocol was also shared with the national program managers.

## Process evaluation

Since the study aims at the feasibility of developing an optimal service delivery model for oral PrEP among the key population of MSM and TGW, the data on process evaluation will be collected as per process assessment parameters (Table 1). This will help in generating evidence on the feasibility of implementing the distribution of oral PrEP medicine to the MSM and TGW population in India. The data will be collected as interview schedule on willingness of participant to continue PrEP after the study is over, his expectation for the cost if this has to be provided by the government under the programme and in-depth qualitative exit interviews of all participants are proposed at termination visit to understand willingness for paid PrEP, median acceptable cost of PrEP, reasons for choosing or declining PrEP, barriers, and facilitators of PrEP use, and challenges faced.

## Primary and secondary endpoints

The study primarily aims at the adherence of MSM and TGW for PrEP medicines, retention in the oral PrEP demonstration study (percentage retention as defined earlier will be calculated for every participant and cumulatively for each study setting), and acceptability that includes an assessment of three points– 1) a proportion of MSM/TGW, who are negative visited the clinic against those reached through demand generation activities, 2) the number of MSM/TGW who are negative, eligible and enrolled against those who were screened and 3) proportion of MSM/TGW who initiated PrEP against those who were enrolled at the baseline.

## Primary and secondary outcomes

### Primary outcomes.

- Retention for the PrEP study—Percentage retention will be calculated for every participant and cumulatively for each study setting.

- Acceptability

- The proportion of HIV negative MSM/TGW visited the clinic against those reached through demand generation activities

- The number of HIV negative MSM/TGW eligible and enrolled for the PrEP study

- The number of HIV negative MSM/TGW who are initiated on PrEP,

- The proportion of PrEP users who tests HIV positive during the study period

- Percentage of the study participants who had completed one year of PrEP use

- Adherence: net pills per day (Actual number of pills taken/Expected number of pills to be taken) x 100

- Detectability of drug in 20% of the blood samples over a period of one-year, self-report

- Side effects and toxicities, including creatinine elevations

- Time-to-event analysis by the occurrence of side effects and discontinuation of the drug.

### Secondary outcomes.

- STI incidence rates (syphilis, NG, CT, and HSV type-2).

**Table 1. PrEP demonstration project among MSM and TGW: Process evaluation plan.**

| | Questions | Data Source | Tools/ Procedure | Timing | Data analysis & Reporting |
|---|---|---|---|---|---|
| **Fidelity** | The extent of implementation? | Clinician, counselor, pharmacist, participants | The semi-structured questionnaire, Observation checklist, and Qualitative tools | Semi-structured questionnaire to be filled at baseline and end-line; Qualitative data collected at first and exit visit | Qualitative data feedback (informal) to the staff one week after the first visit for possible improvements |
| **Dose delivered** | To what extent is PrEP and adherence counseling delivered? | Clinician, counselor, pharmacist, nurse | Self-reported checklist | Every scheduled visit | One week after the visit, issues discovered will be discussed with staff—informal feedback |
| **Dose received** | Was PrEP dispensed properly? Was counseling optimal? Were participants satisfied? | Clinician, counselor, pharmacist, participants | Focus group discussion with the clinical team and community workers. Structured tool/ scale to assess satisfaction with the implementation program | Baseline, midterm, and end line | Final report (summative report) |
| **Reach** | Was the sample size reached at each site? What was the rate of retention? | Data manager/ coordinator; Discussions with outreach staff | Enrollment and follow-up records | Each visit or any additional visit | After each quarter [Formative]. Final report site-wise, the approach of delivery wise [Summative] |
| **Recruitment** | What were the methods of recruitment at each site? | Site outreach team | The outreach team will document all activities | Daily | To be examined weekly to solve problems: will be both formative and summative report |
| **Context** | Barriers and facilitators in implementation | Research staff | FGD | Exit time | Final report (summative report) |

- Number of unprotected sexual acts in the last three months and change in sexual behaviour practices after PrEP

- The proportion of PrEP users willing for paid PrEP (PrEP users willing for paid PrEP/total enrolled participants)

- The median cost of PrEP, participants willing to pay (median cost out of all reported costs borne by the participants)

- Reasons for choosing or declining PrEP

## Clinical safety

We will assess clinical safety by monitoring adverse events during the study period based on the available package insert of PrEP Tablets. Serious adverse events are those that are considered life-threatening or classified as defined per CFR 312.32 guidelines. The severity of Adverse Events (AEs) will be graded according to the DAIDS Table Version 2.1, July 2017 for Grading the Severity of Adult and Pediatric AEs [25].

## Adverse events

Data regarding adverse events- both known and unforeseen- at each visit would be recorded and dealt with. Some of the known adverse events are gastrointestinal side effects and laboratory parameters (reduction in creatinine clearance). There could be any drug-drug interactions. The study clinician will have the discretion to hold oral TDF containing PrEP tablets at any time they feel that continued medication use would be harmful to the participant or interfere with treatment deemed clinically necessary according to the judgment of the clinician.

Clinical or laboratory abnormalities that require follow-up will be documented, and the counsellor or clinician will contact the participant to schedule an interim visit for follow-up and/or repeat laboratory testing. Participation in the PrEP demonstration project could lead to social harms that may include loss of privacy, stigmatization, interference with gainful employment, and coercion. Information regarding social harm will be solicited during follow-up study visits and will be recorded on study documents.

Serious adverse events (SAEs) will be defined per CFR 312.32 guidelines, as AEs occurring at any dose that: result in death, are life-threatening adverse events, require inpatient hospitalization or prolongation of existing hospitalization; result in persistent or significant disability/ incapacity; or congenital anomalies/birth defects. Important medical events that may not result in death, be life-threatening, or require hospitalization may be considered serious when based upon appropriate medical judgment, they may jeopardize the participant or require medical or surgical intervention to prevent one of the outcomes listed above.

AE/ SAE reporting: Monitoring of AE/SAE is critical after initiating PrEP. Monitoring of adverse events will be done by the clinician at every visit. The relationship of all AEs reported by the participants to FTC/TDF will be assessed per the package insert, the investigator's brochure for FTC/TDF, and the clinical judgment of the clinician. Clinical or laboratory abnormalities that require follow-up will be documented. All participants reporting an adverse event will be followed clinically until the occurrence resolves (returns to baseline grade, defined as a grade at enrollment) or stabilizes or an effective referral to local health care providers is accomplished. To ensure external periodic monitoring of adverse events, an Adverse Event Monitoring Committee (AEMC) is constituted. The members of the AEMC include experts from epidemiology, public health, pharmacology, community representatives, statisticians, and program representatives. The Terms of Reference for AEMC are developed during the preparatory phase and the first AEMC meeting was held in November 2020 after the completion of three-monthly follow-up visits of the first 50 enrolled participants. The project steering committee (PSC) independent of investigators and sponsors is formed to look after the annual progress of the study. It consists of program representatives, social scientists, community representatives, subject experts, and experts in MSM and TGW research. Insurance was obtained for the study participants based on the risks involved of side effects due to PrEP Tablet. The Hepatitis B vaccine will be administered to participants in a hospital set up with having casualty set up to manage the participants in case of any adverse reaction to the vaccine.

**Protocol amendments.** Any protocol amendments will be communicated to AEMC and Institutional Ethics Committee-ICMR-National AIDS Research Institute. The amendment will be done in the protocol only after approval from AEMC and Institutional Ethics Committee-ICMR-National AIDS Research Institute.

## Medication adherence

Adherence will be assessed by participant self-report at follow-up visits, and pill count by the pharmacists. Each participant will be given a study diary to mark after he takes the tablet and this will be assessed by the pharmacist and clinician at each follow-up visit. The diary will also act as a reminder for the participants to ensure optimal adherence.

## Data management

Software for data entry has been developed with data entry screens displaying the questions in English and the two local languages—Marathi and Gurumukhi. The study will have a centralized database that will be updated daily. Unique identification codes for screening and enrollment will be assigned to each data collection event. These codes will be used to identify data

collection forms, sample collection vacutainers, and vials as well as any individual-level information collected in field notes. The name of the participants will be collected only on the consent form and it will be accessed only by study staff. The data will be collected by a counsellor, clinician, laboratory technician, and pharmacist at the clinic as well as community-based setting, using tablets/computers. These tablets/computers will be password protected and accessed by the respective designated individuals of the study only. PI and Co-PI will have access to the final dataset of the study. Data will be stored on an external storage device at the study site at a different location and with data management as a data backup.

## Data analysis

There will be two sets considered for analysis. The first one will be the 'Screened Population' which includes all subjects who consent to be screened for the study, regardless of enrollment status. The other set will be 'Enrolled Population' which includes all subjects who consent to participate in the study, enrolled, and received the study drug (PrEP). The retention rate and the associated 95% confidence intervals will be calculated overall and by the study setting. We shall also analyze the data to identify independently associated correlates of retention by adjusting for socio-demographic, psychological, and other cofactors. Similarly, the proportion of HIV Seroconversion among PrEP users and the associated 95% confidence intervals will be calculated. Other key variables will be summarized descriptively across- and by study setting.

Analysis after one year of enrollment initiation will be conducted to identify challenges in study implementation along with an estimation of HIV incidence, PrEP adherence, and assessment of behavioural inhibition and changes in sexual behaviour following PrEP initiation. This analysis would be critical for advocacy for PrEP rollout. Incidence of HIV among PrEP users will be calculated against person time follow up however this will not be used to compare with any baseline data as there is no baseline data available for the incidence of HIV among MSM in India. HIV incidence among participants in the study will be compared to the available baseline incidence of HIV among a similar population in the same location. The factors associated with adherence, acceptability and side effects will be analyzed using bivariable and multivariable analysis independently for both settings with t and chi-square tests.

The data will be assessed independently for the two service delivery models and they will not be compared with each other. Regional and site-based differences will be captured and studied. There is no plan for any subgroup analysis in the current proposal. Missing data will not be replaced by any method of imputation.

## Dissemination and knowledge translation

Study findings will be disseminated through multiple effective means including presentations at conferences and workshops, publishing papers in peer-reviewed journals, reports, technical and policy briefs, and dissemination workshops with stakeholders working with the MSM and TGW community, study participants, and policymakers.

## Study progress

The study enrollments began in January and February 2020 in Pune and Jalandhar sites respectively and follow-up visits are expected to be completed by July 2023. Recruitment and retention were going on as per the timelines however; the SARS-CoV-2 pandemic has wedged community mobilization and referral of MSM and TGW population significantly. There was a significant reduction in recruitment and retention due to lockdown and pandemic challenges. Given this emerging threat to recruitment, virtual meetings and participatory approach are the innovative approaches implemented for community mobilization, and their outcome would

be evaluated after a few months. The study is closed for recruitment on 31 July 2022. The project would require additional 12–15 months extension.

## Discussion

HIV prevention is the key to ending AIDS by 2030. India has shown good progress in increasing treatment but we are lagging in prevention. National AIDS Control Organization has played a pivotal role in the prevention and control of HIV/AIDS seen as a reduction in prevalence to 0.22% [1, 4] However new infections among key populations continue to be seen. Epidemics concentrated in key populations are also expanding to other at-risk populations and the only way to contain the epidemic is to prevent the emergence of new infections by using proven biomedical and behavioural HIV prevention options.

The efficacy of the PrEP is established through various clinical trials globally [7, 26, 27]. However, all trials have reported issues that could affect the effectiveness. A person's decision-making for health-seeking is governed by individual and shared decisions and community norms and expectations. It also gets influenced by the health system environment, provider-related characteristics, and behaviours. Thus, the nature of care-seeking is not homogenous depending on cognitive and non-cognitive factors that call for contextual modalities to be dealt with. These contextual factors would undermine the success of the real-world oral PrEP rollout program among the MSM and TGW populations in India. The feasibility in a real-life setting, acceptable delivery strategies, facilitators, and barriers to the uptake of oral PrEP among the MSM and TGW population in India need to be understood and informed about to the program [27].

Currently, there are no program guidelines for PrEP among the high-risk population in India. There is a need to generate in-country evidence and learnings related to PrEP, to determine the feasibility and acceptability of PrEP in an Indian context, and to identify the most effective strategies for roll-out. This prospective cohort demonstration project aims to assess the uptake, acceptability, safety, and feasibility of daily oral PrEP among men reporting sex with men and transgender women at the clinic and community-based settings for one year.

To the best of our knowledge, this is the first Indian demonstration project aiming at studying the feasibility of providing orally administered PrEP among the MSM and TGW population in India. By definition, demonstration projects are an intermediary step between initial efficacy research and full-scale implementation. They "demonstrate" that the intervention that is proven to be effective in controlled clinical conditions in clinical trials can be implemented in settings reflective of real-life health system challenges. The uniqueness of this project is that it aims at generating evidence on process evaluation for administering oral PrEP to MSM and transgender women population in the Indian socio-cultural and system context.

The clinic-based model is considered the most ideal for delivering PrEP as PrEP requires clinical monitoring for creatinine clearance before the initiation and continued periodic monitoring after initiation. Anti-retroviral treatment (ART) centres under the national program that provide ART to HIV-infected individuals are equipped with manpower and laboratory backup. These can serve as the best service delivery option however they are questioned by perceived stigma by the MSM and TGW persons. There have been concerns raised by the community about service delivery, long waiting periods, anticipated stigma, and discrimination by healthcare providers [28]. It is thought that the perceived stigma would be less in the community-based model as it will be implemented by service providers from the community/peers. A community-based approach for oral PrEP delivery to MSM and TGW populations could increase and sustain access to oral PrEP, in addition to a facility-based delivery model however, evidence is limited on monitoring needed after PrEP initiation.

The US PrEP Demonstration Project has reported a high level of interest in PrEP when offered in STI and community health clinics [29]. There is no evidence in the country on its actual feasibility and effect on adherence and monitoring in this model. Hence this study will help the policymakers and program in designing a delivery system for oral PrEP for MSM and TGW populations.

The key factors that influence the willingness or use of PrEP include acceptability, accessibility, side effects, and cost of biomedical prevention methods [28]. The acceptability of PrEP among MSM and TGW population has been found to be high in various clinical trials [7, 28, 30, 31] however, there were limited data on the preferred mode of PrEP dosing, service delivery sites, monitoring, and implementation. This demonstration project will provide data on the acceptability of the community and the intricacies of the two service delivery models for oral PrEP among MSM and TGW persons in the Indian program setting. The learnings would be useful for strengthening both the service delivery settings.

With the COVID pandemic, there has been a fear in the community that is reflected in the change in high-risk behaviour. Risk behaviours have become more covert and probably more unsafe and unprotected. Therefore, interruptions to control programs could result in major setbacks, compounding the direct impact of COVID-19 [32]. There is an urgent need for political will to consider PrEP for MSM and TGW population for HIV prevention and containing transmission.

## Strengths and limitations of the study

The strength of this study is the sample size which includes 300 participants at each study site making it 600 at both sites including 20% TGW persons. Since the sample size is based on guidance from WHO on pre-exposure prophylaxis (PrEP) demonstration projects, it will allow the investigators for international comparisons. Methodologies for determining adherence are a critical discussion point, especially the frequency and periodicity of blood sample collection. The current study has been proposed considering the methodologies from the program perspective. The proposed process evaluation forms another strength of this study as it will provide evidence on actual implementation issues like fidelity, dose delivered, received, cost, reach, and recruitment. This will help in designing operational guidelines for PrEP delivery. This study would also help in strategizing toward enhancing the resilience of the health system that needs to cope with sudden pandemics/ outbreaks. The single-arm and two study sites will limit the generalizability of results and another limitation is thought to be a potential threat to achieving targets due to the COVID-19 pandemic. Participants with HBV are excluded to avoid a flare of HBV infection because of limited access to PrEP for the participants after the termination from the study. Desai et al predict an increase in HCV infection among HIV-negative MSM who access pre-exposure prophylaxis (PrEP) as they reported sexual behaviour that could place them at high risk of HCV. Therefore, post-participation in the PrEP demonstration, participants may not continue to take PrEP which could lead to HCV flare up. To prevent the potential for flare up and disinhibition in terms of drug injection practices, HCV-positive individuals will be excluded from this demonstration project [33, 34]. This is a limitation of this demonstration project however screening for HBV and HCV are linked to care will be ensured. One of the exclusion criteria includes participants who have any condition that, based on the opinion of the investigator or designee, would preclude the provision of informed consent; make participation in the project unsafe; complicate the interpretation of outcome data, or otherwise interfere with achieving the project objectives. This could be a limitation however it will be the rarest of the rare condition. Any such exclusion will be discussed

with the PI, Study chair and clinician for consensus. Such exclusions if done will be well documented.

## Conclusions

This demonstration project will generate evidence on PrEP that is specific to the cultural contexts and health systems in India to support and inform governments' decisions in incorporating PrEP as a public health policy. The most useful evidence drawn from this demonstration project will be grounded in the reality of existing the National Program and COVID-19 pandemic.

## Ethics approval and consent to participate

The study is approved by the Institutional Ethics Committees of ICMR-National AIDS Research Institute (EC Protocol Number: NARI EC/2017-07), HIV/AIDS India Alliance, and also presented to the Central Drugs Standard Control Organization (CDSCO). All possible endeavors will be undertaken to respect and protect the rights and dignity of participants at all stages of the study. Each participant will be given thorough information on the nature and scope of the research to be conducted. All study participants will be requested to provide voluntary written informed consent before participating in the study in a culturally sensitive manner and in a participant's language of preference (i.e. Marathi, Hindi, English, and Punjabi/ Gurmukhi).

To minimize the breach of participant's privacy and confidentiality, we will secure all informed consent forms and forms containing identifying information in a locked file cabinet. We will ensure all study-related documents besides the consent forms are devoid of identifying information. All data will be coded by a participant identification number (PID).

## Acknowledgments

Our sincere thanks to the study respondents for their time and for sharing information for this study. We acknowledge the efforts of the study staff in supporting the data collection activities in this study. We thank Dr. Kenneth Mayer, Professor, Department of Global Health and Population, Harvard Medical School, and Dr. Ioannis (Yannis) Hodges-Mameletzis HIV/PrEP advisor at World Health Organization Ukraine Country Office

## Author Contributions

**Conceptualization:** Seema Sahay.

**Funding acquisition:** Seema Sahay.

**Methodology:** Seema Sahay, Sampada Bangar, Nomita Chandhiok.

**Supervision:** Seema Sahay, Sampada Bangar, Nomita Chandhiok.

**Writing – original draft:** Seema Sahay, Sampada Bangar, Nomita Chandhiok.

**Writing – review & editing:** Seema Sahay, Sampada Bangar, Nomita Chandhiok.

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
