## [Decision Letter · Decision Letter 0]

28 Mar 2023

PONE-D-22-31093Demonstration project of Oral TDF-containing PrEP, administered, once-daily orally to men having sex with men (MSM) and transgender women (TGW) in India: Study protocolPLOS ONE

Dear Dr. Sahay,

Thank you for submitting your manuscript to PLOS ONE. After careful consideration, we feel that it has merit but does not fully meet PLOS ONE’s publication criteria as it currently stands. Therefore, we invite you to submit a revised version of the manuscript that addresses the points raised during the review process.

I would like to sincerely apologise for the delay you have incurred with your submission. It has been exceptionally difficult to secure reviewers to evaluate your study. We have now received two completed reviews; the comments are available below. The reviewers have raised significant scientific concerns about the study that need to be addressed in a revision.

Please revise the manuscript to address all the reviewer's comments in a point-by-point response in order to ensure it is meeting the journal's publication criteria. Please note that the revised manuscript will need to undergo further review, we thus cannot at this point anticipate the outcome of the evaluation process.

We look forward to receiving your revised manuscript.

Kind regards,

Miquel Vall-llosera Camps

Senior Editor

PLOS ONE

Journal Requirements:

This study is funded by Indian Council of Medical Research, Government of India (Grant Number: 5/7/2/2016-RBMH).

Reviewers' comments:

Reviewer's Responses to Questions

**Comments to the Author**

1. Does the manuscript provide a valid rationale for the proposed study, with clearly identified and justified research questions?

Reviewer #1: Partly

Reviewer #2: Yes

2. Is the protocol technically sound and planned in a manner that will lead to a meaningful outcome and allow testing the stated hypotheses?

Reviewer #1: Partly

Reviewer #2: Yes

3. Is the methodology feasible and described in sufficient detail to allow the work to be replicable?

Reviewer #1: No

Reviewer #2: Yes

4. Have the authors described where all data underlying the findings will be made available when the study is complete?

Reviewer #1: Yes

Reviewer #2: No

5. Is the manuscript presented in an intelligible fashion and written in standard English?

Reviewer #1: No

Reviewer #2: Yes

6. Review Comments to the Author

You may also provide optional suggestions and comments to authors that they might find helpful in planning their study.

Reviewer #1: I uploaded my review as a PDF file attachment which explains my answers to the questions above and provides comments about issues authors must address before this protocol can be accepted for publication.

Reviewer #2: Author to enumerate the rationale for enrollment of only 20% of TGW study participants versus MSM study participants in the study design.

7. PLOS authors have the option to publish the peer review history of their article (what does this mean?). If published, this will include your full peer review and any attached files.

Reviewer #1: No

Reviewer #2: No

---

## [Author Response · Author response to Decision Letter 0]

22 May 2023

Pointwise response to reviewer’s comments

There are many instances throughout the manuscript where data is provided, or assertions are made based upon prior research where the source(s) are not cited or are cited incorrectly.

Response: Thank you for your comment. We have updated the references as per your instructions. 

The study purpose/objectives/aims, where described at different points in the manuscript are inconsistent.

You should use the WHO “PrEP demonstration projects. A framework for country level protocol development” as a guide, and focus on describing the overarching purpose of the study ‐ i.e. to evaluate the feasibility and potential impact of implementing HIV PrEP as part of combination prevention for HIV among MSM and TGW in India. Then describe the factors you will assess to evaluate that main purpose ‐ i.e. by assessing adherence to PrEP and the incidence of HIV infection among study participants compared to the incidence of HIV among a similar population in the same area. You can then further elaborate on specific indicators/measures related to adherence such as facilitators/barriers to adherence, willingness to pay for PrEP, retention in the study, etc. as well as your secondary outcomes of interest that you plan to assess in this study. 

Response: Thank you for your comments. The study was designed following the WHO’s PrEP demonstration projects: A framework for country level protocol development. The reference is added in the manuscript in objectives section. The primary goal of the study is added as to evaluate the feasibility and potential impact of implementing HIV PrEP as part of combination prevention for HIV among MSM and TGW in India. It has strengthened the manuscript and added clarity to it. 

Line no: 128-130

Page no: 7

There are multiple instances throughout the manuscript where the full word/phrase/term that an abbreviation represents does not precede the first use of that abbreviation that should be corrected. This should be done for the first use of an abbreviation in the abstract as well as its first use in the body of the manuscript. 

Response: Thank you for your comment. We have added the full word/phrase/term that an abbreviation represents at its first use in the manuscript. 

The full word/phrase/term that an abbreviation (HIV in this case) represents should be included prior to that abbreviation first use in the text 

Response: Thank you for your guidance. We have revised the manuscript and expanded all the abbreviations at its first use in the manuscript. 

Line no: 17, 45

Page no: 2, 4

Should include full term for “MSM” before the abbreviation 

Response: Thank you for your guidance. We have revised the manuscript and expanded “MSM” in the manuscript. 

Line no: 18

Page no: 2

The full word/phrase/term that an abbreviation (TDF in this case) represents should be included prior to that abbreviation first use in the text 

Response: Thank you for your comment. We have included the full form of TDF at its first use in the manuscript. 

Line no: 24

Page no: 2

You describe your primary outcomes of interest here but do not include “the proportion of PrEP users who tests HIV positive during the study period” which is listed as a primary outcome later in the manuscript (line 271). 

Response: Thank you for your guidance. We have mentioned breakthrough infections while on PrEP in the primary outcomes as per the suggestion of the reviewer in the abstract 

Line no: 31

Page no: 2

You also describe “sexual behavioral changes with PrEP use” as a primary outcome of interest here, but this is not listed as a primary or specifically as a secondary outcome elsewhere in the manuscript. 

Response: It is mentioned in the manuscript in primary outcomes at 

Line no: 336-337

Page no: 15

As this study attempting to assess the feasibility of PrEP in the populations at highest risk of HIV infection in India, which you postulate are the MSM and TGW populations, you should include data on the incidence of HIV infection in these populations in the background section that supports that these populations are in fact the highest risk populations in India for new HIV infection. For example, define how many of the approximately 69,000 new HIV infections in line 48 occurred amongst those that identify as MSM or TGW. 

Response: Thank you for your suggestion. We have added sentence regarding incidence in the revised manuscript. 

Line no 51-52

Page no: 4

Again on its first use in the body of the manuscript 

Response: Thank you for your guidance. We have added full form of HIV first line of manuscript. 

Line no 45

Page no: 4

India has demonstrated considerable success with a 37% decline in new HIV infections since the year 2010, yet around 69,000 new infections occurred in 2019” 

Response: Thank you for your comment. We have added reference to it. 

Line no: 49

Page no: 4

This data describing increased likelihood for HIV infection among the MSM population does not contribute much to the rational for this study – you aren’t targeting any of these specific subpopulations of MSM 

Response: Thank you for your guidance. We agree with the reviewer and we have removed the reference. 

While there is data to support male circumcision as effective in the prevention of HIV among heterosexual men, this data does not support prevention solely from insertive vaginal intercourse (the studies didn’t obtain any data on the specific types of sexually activity with the participants’ partners)

Response: Thank you for your guidance. We agree with the reviewer’s comments, and we have removed the reference.

While VMMC may not be culturally acceptable in India, this is irrelevant as there is no evidence that VMMC is effective in the prevention of HIV infection among your target population (MSM and TGW), so this isn’t good justification to pursue alternative prevention strategies in your context. 

Response: We agree with the reviewer’s comments, and we have removed the reference.

Per the most recent update of the clinical practice guideline for PrEP for the prevention of HIV in the US, the US CDC recommends offering PrEP to adults and adolescents whose sexual activity or injection drug use practices place them at substantial ongoing risk of HIV acquisition.

Response: We agree with the reviewer. We have revised the sentence and added the recent reference to support this. 

Line no 76-79

Page no 5

Given the updated 2015 PrEP recommendations from the WHO which includes using PrEP as part of a combination prevention package, the 2014 recommendation is rather irrelevant. 

Response: Thank you for your suggestion. We have revised the sentence and added the recent reference to support this. 

Line no. 85-88

Page no 5

UNAIDS recommendations were not discussed previously, nor otherwise referenced, so it is unclear why they are mentioned here 

Response: Thank you for pointing it out. We agree with reviewer. We have removed UNAIDS from the sentence 

A gap between efficacy and effectiveness is predicted as many clinical trial researchers cited adherence as a major challenge for efficacy” have no sources cited. 

Response: Thank you for your suggestion. We have added the references. 

Line no 106

Page no 6

Should specify that the “real world” setting in this case is in India. 

Response: Thank you for your suggestions. We have revised as suggested. 

Line no: 109

Page no: 6

You need to provide a description of the community‐based setting (none included) and more detail on the facility‐based setting such as specific location/facility, why that location was chosen, if it is embedded in an existing clinic (and if so, what type of clinic/other services offered) or was created for the sole purpose of this trial 

Response: Thank you for your suggestion. We have added the description for community and facility-based settings in the revised manuscript. 

Line no: 147-155

Page no: 8

Why did you select these states for your study sites? Are these the states with the highest HIV prevalence (although incidence would be a more appropriate measure for this study) in India and is this the most recently available data? You report that the HSS is conducted yearly earlier in the background section. 

Response: Thank you for your suggestion. These states were selected as they had higher HIV prevalence among MSM than the national prevalence reported at the time of protocol submission to the funding agency. We have revised the sentence to include reason for selection of these states. 

Line no: 144-145

Page no: 8

What is the reason that you decided to establish the community‐based vs the facility-based setting in one location vs the other? This should be discussed. 

Response: Thank you for your suggestion. We decided to establish the community‐based and the facility-based setting in different locations to avoid spill- over of the community between two settings. We wanted to include more sites (both facility and community based) for a more representative sample however were limited because of funding issues. The study sample size is not powered to compare the two different settings and therefore the analysis does not include any comparison between the delivery models.

Why do you expect differences in stigma in between the two locations based on geographic location? This should be discussed.

Response: Thank you for pointing this out. We have revised as suggested. 

Line no: 158-160

Page : 8

How did you determine the need for/sufficiency of 20% of the study population being TGW and how do you plan to ensure that 20% of your participants will be TGW? This should be discussed. 

Response: Thank you for your TGW population are difficult to reach especially for PrEP because they require endorsement from their community leaders/Gurus however still an attempt would be made to enroll 20% TGW population of the total sample size across both the sites. TGW population will be enrolled based on their self-report. 

The wording here suggests that your criteria defining substantial risk for HIV infection are from the WHO PrEP demonstration projects guidelines – they are not as this guideline does not specify any specific criteria to define substantial ongoing risk for any population. You should specify why you chose these specific criteria to identify participants at substantial ongoing risk for HIV infection and/or what guideline/reference you obtained them from. Why are you not including persons with an HIV positive sex partner, persons using injection drugs whose practices place them at substantial ongoing risk of HIV acquisition? 

Response: Thank you for your suggestion. The specific criteria to define substantial risk are taken from the WHO’s implementation tool for pre-exposure prophylaxis (‎PrEP)‎ of HIV infection: module 1: clinical. We have revised the manuscript and added this reference. 

Line no: 178

Page no: 9

HIV positive sex partner, persons using injection drugs whose practices place them at substantial ongoing risk of HIV acquisition are not exclusion criteria so if they fulfil any of the substantial risks then they are eligible for enrollment.

The full word/phrase/term that an abbreviation (HBV and HCV in this case) represents should be included prior to that abbreviation first use in the text 

Response: Thank you for your guidance. We have added full form of HBV and HCV at its first use in the manuscript. 

Line no: 176, 184-185

Page no: 9

What are the contraindications to oral TDF containing PrEP medications used in your study? This should be discussed. 

Response: Thank you for your suggestion. We have added the contraindications in the revised manuscript. 

Line no: 185-191

Page no: 9

The full word/phrase/term that an abbreviation (CBOs/NGOs in this case) represents should be included prior to that abbreviation first use in the text 

Response: Thank you for your guidance. We have added full form of CBOs and NGOs at its first use in the manuscript. 

Line no 204-205

Page no: 10

You have not included any measures to assess participants for acute HIV infection during screening such as screening for signs or symptoms suggestive of acute HIV infection within the previous four weeks or high‐risk exposure within four weeks of starting PrEP. This is an extremely important step in the assessment of patients for initiation of PrEP to prevent starting PrEP in someone with acute HIV infection. 

Response: Thank you for your suggestion. We have revised the manuscript and added measures to assess participants with acute infection. This has strengthened the manuscript. 

Line no: 226-230

Page no 11

The specific tests to be used in this HIV screening process for your study need to be described i.e., manufacturer and name of test along with what type of test it is such as “the Abbot DETERMINE™ HIV‐1/2 AG/AB COMBO, a 4th generation HIV antigen/antibody test that detects both HIV‐1/2 antibodies and free HIV‐1 p24 antigen”

Response: Thank you for your suggestion. We have added the specific tests used for HIV screening in the revised manuscript. 

Line no: 230-231

Page no: 11

Within what timeframe will participants undergo the enrollment visit after screening? This should be discussed. 

Response: Thank you for your comment. We have specified the duration at 

Line no: 232

Page no: 11

Recommend removing “and” from “All enrolled and HBV‐non immune” to clarify that it will be the HBV non‐immune participants enrolled in the study who will be offered HBV vaccination 

Response: Thank you. We have revised as suggested. 

Line no: 246

Page no: 11

frequency of HIV screening, serum creatinine measurement (including any plans for increased frequency of measurement for those with risk factors for renal disease such as hypertension, diabetes, etc.), and questionnaires/surveys to obtain data on sexual behaviors, risk perception, facilitators/barriers to adherence, use of other HIV prevention options, behavioral/psychologic health to measure your study objectives should be discussed here. Screening for symptoms suggestive of acute HIV infection should also be included at study visits. 

Response: Thank you for your suggestion. We have revised the manuscript to include frequency of data collection. 

Line no: 238-244

Page no: 11-12

We would like to mention that we have also mentioned schedule of events in the manuscript which covers all laboratory investigations. 

Line no: 282 

Page no: 13

Figure 2 (uploaded separately)

As written it seems that side effects will be assessed, and adherence counseling will be done only during the first 6 visits – this should be done at every study visit 

Response: Thank you for your comment. The adherence counseling is proposed to be done at each visit. We have revised the sentence to reflect the same. 

Line no : 254-255

Page no: 12

You need to describe how these samples will be managed/stored for future Tenofovir drug level monitoring as well as the specific test you will use to test Tenofovir drug levels. 

Response: We have added the information regarding TDF monitoring and testing. Thank you.

Line no 257-260

Page no 12

Why are you not including C. trachomatis and N. gonorrhea PCR testing on pharyngeal and rectal swab samples to screen for pharyngeal/rectal gonorrhea and rectal chlamydia?

Response: Thank you for your suggestion. We have validated C. trachomatis and N. gonorrhea PCR testing using urine sample at our laboratory. With our past experience, MSM and TGW are more comfortable in giving urine sample and most of the times they refuse rectal/pharyngeal sample collection. 

There is generally no role for serologic screening for HSV in asymptomatic persons as HSV‐2 serologic tests have a low specificity and a high false‐positive rate when used for screening asymptomatic individuals, and there is no specific treatment warranted for an asymptomatic individual. What is your rationale for including screening for HSV in your study? 

Response: Thank you for your comment. HSV2 is sexually transmitted infection and test was included to understand baseline prevalence of HSV irrespective of symptoms or no symptoms. Herpes Simplex infection is considered a surrogate marker for high-risk sexual behavior. We agree that there is no cure for Herpes infection however appropriate counseling to asymptomatic but HSV2 positive cases can prevent the transmission of HSV (genital) to other organs if touched without washing their hands after touching sore or liquid from sore or to their partners during oral or genital sex. Considering increased possibilities of oral and anal sex among this population this test was included in the project. 

What specific type of HSV testing will be performed for a participant with an ulcer suspected to be caused by HSV? This should be discussed. 

Response: Thank you for your comment. We have included the HSV test. 

Line no: 267

Page no 12

It would be ideal to include this in determining eligibility to help minimize study dropouts Response: Thank you for your comment. Being HSV 2 positive is not an exclusion criterion for PrEP as per the WHO PrEP demonstration project guidelines.

Line no: 194-200

Page no: 9

How many days’ supply of PrEP will participants receive and at what visits will they receive it? This should be discussed. 

Response: Thank you for your comment. We have added the information in study medication section. 

Line no: 289-291

Page no: 13-14

You should specify the dose of Emtricitabine in each tablet. 

Response: Thank you for your comment. We have revised as per suggestion. 

Line no: 287

Page no: 13

You state here that participants will receive specifically Tenofovir Disoproxil Fumarate‐Emtricitabine for PrEP as the study medication, but elsewhere throughout the manuscript you refer to imply that either Tenofovir Disoproxil Fumarate‐Emtricitabine or Tenofovir Disoproxil Fumarate‐Lamivudine may be used in the study (lines 162, 167). If participants will be receiving only Tenofovir Disoproxil Fumarate‐ Emtricitabine as the study medication for PrEP in this study, then references to Tenofovir Disoproxil Fumarate‐Lamivudine should not be included other than for background information. 

Response: We have excluded as per your suggestion. Thank you. 

Your method for determining acceptability is unclear as written. Are you defining acceptability as the proportion of persons who actually enroll in the study and initiate PrEP out of all the persons determined to be eligible and offered enrollment in the study. Again, it is unclear how you are defining/determining acceptance based on this description. 

Response: Thank you for this question. We have described in detail three points of acceptability assessment.

Line no: 312-316

Page no: 14-15

How are you determining the number of unprotected sexual acts in the last three months (you provide no description anywhere in the manuscript of a study questionnaire/survey to obtain this data at follow‐up visits)? Is this the sole measure for your previously stated study objective of evaluating change in sexual behaviors in response to PrEP? Number, type, and new sex partners as well as type of sex acts would provide additional valuable insight regarding changes in sexual behavior. 

Response: Thank you for rightly pointing out the missing information. We have described the focus and frequency of sexual behavior data collection in the revised manuscript. 

Line no: 238-241

Page no: 11-12

How are you determining the number of study participants exhibiting anxiety, depression, and suicidal ideation (you provide no description anywhere in the manuscript of a study questionnaire/survey to obtain this data at follow‐up visits)? 

Response: Thank you for your suggestion. We are not using any scale therefore we have removed this secondary outcome after your review. We have described data collection on emotional distress, abusive relations and suicidal ideations which will be part of the semi-structured questionnaire. We are very thankful for your guidance. 

Line – 242-244

Page no: 12

How are you determining the proportion of PrEP users willing for paid PrEP (you provide no description anywhere in the manuscript of a study questionnaire/survey to obtain this data at follow up visits)? How are you determining the median cost of PrEP, participants willing to pay (you provide no description anywhere in the manuscript of a study questionnaire/survey to obtain this data at follow‐up visits)? How are you determining the reasons for choosing or declining PrEP (you provide no description anywhere in the manuscript on how this data will be obtained when eligible participants are offered enrollment or a study questionnaire/survey to obtain this data at follow‐up visits)? 

Response: Thank you for your suggestion. This data will be collected all participants at termination visit. We have revised the manuscript to include this. 

Line no: 303-308

Page no: 14 

How will you determine the number of successful partner referrals (you provide no description anywhere in the manuscript on how this data will be obtained/tracked)?

Response: Thank you for your suggestion. Owing to confidentiality reasons as communicated by key community leaders, we are not collecting this data. Accordingly, this outcome is removed from the manuscript. We are really thankful for this critical review. 

Here you state, “Serious adverse events are those that are considered life‐threatening”, but later in lines 307‐310, you describe multiple different scenarios that define a serious adverse event. This discrepancy needs corrected. 

Response: Thank you for your guidance. We have corrected the discrepancy. 

Line no: 349

Page no: 16

What are the statistical tests you will use to perform these analyses? This should be discussed. 

Response: Thanks for your comment. The data will not be compared between two settings hence all data will be calculated as percentages along with associated confidence intervals. We have added description of analysis in the section. 

Line no 430-432

Page no 19

You state that no baseline data for HIV incidence among MSM in India is available, but the “National AIDS Control Organization & ICMR‐National Institute of Medical Statistics. India HIV Estimates 2019”, which you cite in your manuscript, contains data on HIV incidence among MSM. Comparing the HIV incidence among participants in your study to the incidence of HIV among a similar population in the same location would be very informative to determine the potential impact of an oral PrEP program on the prevention of HIV and is a primary outcome of interest suggested by the WHO “PrEP demonstration projects. A framework for country level protocol development”. 

Response: Thank you for your suggestion. We have included the incidence estimates for Punjab from the reference given while for Maharashtra another reference is included now. Data analysis section is also revised accordingly.

Line no: 51-52, 428-430

Page no: 4, 19

Why are you not comparing the two service delivery models? 

Response: Thank you for your comment. The objective of the study is demonstrate feasibility of PrEP delivery in any setting. Therefore, the sample size is not powered for this comparison hence no comparison is planned. 

“National AIDS Control Organization has played a pivotal role in the prevention and control of HIV/AIDS seen as a reduction in prevalence to 0.22%” the source is incorrectly attributed to reference number 3 (The NACO 2014‐2015 Annual Report – national HIV prevalence in India estimates at 0.35% in this report) when it is actually from reference number 1 (the NACO and ICMR‐National Institute of Medical Statistics India HIV Estimates 2019 Report).

Response: Thank you for your suggestion. We have revised accordingly. 

Line no: 456

Page no: 20 

Multi‐center is not an appropriate description of this study as your two sites have different service delivery methods (community‐based vs facility‐based). 

Response: Thank you for your suggestion. We have removed the word as suggested. 

Line no: 473

Page no: 21

How does excluding persons with HCV infection prevent behavioral disinhibition among PrEP participants and prevent any future HCV positivity in this study? 

Response: Thank you for bringing this to our notice. We have clarified our stand to the best of our ability. 

Line no: 530-534

Page no: 23

---

## [Editor Report · Decision Letter 1]

6 Jun 2023

Demonstration project of Oral TDF-containing PrEP, administered, once-daily orally to men having sex with men (MSM) and transgender women (TGW) in India: Study protocol

PONE-D-22-31093R1

Dear Dr. Sandy,

We’re pleased to inform you that your manuscript has been judged scientifically suitable for publication and will be formally accepted for publication once it meets all outstanding technical requirements.

Kind regards,

Sandra Ann Springer, M.D.

Academic Editor

PLOS ONE

Additional Editor Comments (optional):

The authors have improved their manuscript through their revisions.
---

## [Editor Report · Acceptance letter]

13 Jun 2023

PONE-D-22-31093R1 

Demonstration project of Oral TDF-containing PrEP, administered, once-daily orally to men having sex with men (MSM) and transgender women (TGW) in India: Study protocol 

Dear Dr. Sahay:

I'm pleased to inform you that your manuscript has been deemed suitable for publication in PLOS ONE. Congratulations! Your manuscript is now with our production department. 

Kind regards, 

on behalf of

Dr. Sandra Ann Springer 

Academic Editor

PLOS ONE